# Comparing Factors Related to Any Conventional Cigarette Smokers, Exclusive New Alternative Product Users, and Non-Users among Japanese Youth: A Nationwide Survey

**DOI:** 10.3390/ijerph17093128

**Published:** 2020-04-30

**Authors:** Yuki Kuwabara, Aya Kinjo, Maya Fujii, Aya Imamoto, Yoneatsu Osaki, Ann McNeill, Nicholas Beckley-Hoelscher

**Affiliations:** 1Division of Environmental and Preventive Medicine, Faculty of Medicine, Tottori University, Tottori 683-8503, Japan; kinjo_aya@tottori-u.ac.jp (A.K.); maya15@live.jp (M.F.); aya.immt@gmail.com (A.I.); yoneatsu@tottori-u.ac.jp (Y.O.); 2National Addiction Centre, Institute of Psychiatry, Psychology & Neuroscience (IoPPN), King’s College London, London SE5 8BB, UK; ann.mcneill@kcl.ac.uk; 3School of Population Health and Environmental Sciences, King’s College London, London SE1 1UL, UK; nicholas.beckley-hoelscher@kcl.ac.uk

**Keywords:** cigarette smoking, e-cigarettes, tobacco use, adolescents, smoking, heat-not-burn tobacco, heated tobacco product, Japan, tobacco control policies, WHO Framework Convention on Tobacco Control (FCTC), noncombustible tobacco/nicotine products, harm reduction

## Abstract

The impact of heated-tobacco-products (HTPs) and electronic cigarettes (e-cigarettes) on youth is a controversial public health issue, as it is unknown whether alternative products result in more youth using such products or smoking. In Japan, e-cigarettes with nicotine are prohibited, but e-cigarettes without nicotine are available. HTPs are marketed as tobacco products. Within this unique context, we aimed to compare any conventional cigarette smokers (including those who also used alternative products) with exclusive users of alternative products and examine factors relating to their use in Japan. In 2017, 22,275 students in grades 7–9 (age 12–15) and 42,142 in grades 10–12 (age 15–18) nationwide were surveyed. Overall, 1.8% were current users of any of the three products over the last month. Multivariable analysis revealed that risk factors for alternative product use were the same as those for cigarette use. Among all users, exclusive new product users were more likely to participate in club activities and intend to continue to higher education; any conventional cigarette users (including those who also used alternative products) were more likely to be exposed to secondhand smoke at home and to drink alcohol. Reducing adult smoking and disseminating health education remain relevant as strategies for preventing adolescents’ future tobacco use.

## 1. Introduction

Tobacco control presents a crucial public health challenge worldwide. A wide range of health problems are attributable to tobacco use, including not only non-communicable diseases, but also perinatal problems and impaired physical and mental development [1]. In fact, premature death due to tobacco use is more preventable than deaths caused by any other drugs [2]. Moreover, the younger people are when they start smoking, the more likely they are to continue smoking [3], making them susceptible to well established smoking-related diseases including cancer, cardiovascular disease and respiratory diseases. The World Health Organization (WHO) has taken the initiative in promoting global tobacco control. Since 2008, the WHO Framework Convention on Tobacco Control (FCTC) has been endorsing practical and cost-effective ways to reduce tobacco demand worldwide using the acronym MPOWER, standing for Monitor (use and prevention), Protect (people from smoke), Offer (help in quitting), Warn (about dangers), Enforce (bans on advertising, promotions and sponsorship) and Raise (taxes on tobacco products).

Smoking cigarettes comprises one of the biggest risk factors for death and disease in Japan, as indicated by a study that estimated the contribution of several risk factors to disability-adjusted life years (DALYs) in Japan from the findings of the Global Burden of Disease (GBD) 2010 project [4]. The Institute for Health Metrics and Evaluation (IHME), a research center dedicating to measuring health problems affecting populations worldwide, reported that cigarette smoking was the dominant risk for death and disability combined in Japan for all years from 2007 through 2017 [5]. Due to the seriousness of the problems associated with smoking, the Ministry of Health, Labor, and Welfare (MHLW) has been tackling issues of tobacco control. In 2000, the national health promotion campaign ‘Healthy Japan 21’ was launched, which included dissemination of knowledge about the health effects of smoking, encouraging youth to stop smoking, calling for the separation of smoking areas and launching a smoking cessation program. In 2003, the Health Promotion Law mandated the management of public facilities to prevent public exposure to secondhand smoke. The law promoted the creation of smoke-free spaces and smoke-free regulation by local authorities. Since 2013, Healthy Japan 21 (second term) has been in effect. The tobacco control measure included four main objectives: (1) decrease the smoking rate for adults from 19.5% in 2010 to 12% in 11 years; (2) eliminate smoking among adolescents and young adults; (3) stop women from smoking during pregnancy; and (4) decrease the occurrence of secondhand smoke and eliminate smoke exposure in all administrative and medical institutions. These political measures added to the already-existing nationwide tobacco controls and smoking prevalence has been decreasing for both adults and adolescents over several decades [6], yet tobacco regulation in Japan still has not met all the FCTC’s recommendations [7].

Meanwhile, new types of alternative nicotine delivery products such as e-cigarettes and various heated tobacco products (HTPs, also called heat-not-burn tobacco products) have emerged in the tobacco market. The e-cigarettes market has rapidly expanded according to a report from the WHO [8]. This matches the findings of a separate study that reported on the prevalence of these new products in several parts of the world [9]. Additionally, the U.S. Center for Disease Control recently reported the trial of e-cigarettes (people ‘trying’ the product) exceeds that of conventional cigarettes in the US [10]. A study from Japan also indicated an increase in the trial of new alternative products [11].

Recently, the U.S. Food and Drug Administration (FDA) permitted the sale of IQOS—an HTP that generates a nicotine aerosol compound—with restrictions on how it can be marketed [12]. While tobacco industries advertise that the tobacco used in HTPs is cleaner, public health experts are discussing the challenges and opportunities created by these new products.

Some of the challenges involve how to categorize and regulate the products, and regulation varies across different countries. Notably, Japan is unique in that national law prohibits selling e-cigarettes containing nicotine, but HTPs are sold as legal tobacco products. In terms of regulation, e-cigarettes without nicotine are available to everyone in Japan, but the purchase and use of HTPs are prohibited among people aged under twenty years old. The age requirement for cigarettes is also twenty years old.

There is a significant and controversial public health concern as to whether the novel products attract susceptible young non-users to initiate use of e-cigarettes or HTPs and then go on to smoking [13]. Previous studies have shown that some youth who are otherwise at low risk for smoking cigarettes, and therefore at low risk for using nicotine, are attracted to using e-cigarettes [14] and later begin to smoke conventional cigarettes [15]. Moreover, to consider the overall population impact of HTPs, more evidence is necessary to inform discussion of the likelihood of adolescents who are not tobacco users or who are former tobacco users adopting the use of nicotine with the new products [16]. On the other hand, if the novel products are confined to youth who are already likely to become smokers, or who use them to stop smoking, they may represent an opportunity to reduce the number of adolescents who are harmed by the effects of combustible cigarettes [17].

The purpose of this study was to compare the background of conventional cigarette smokers with exclusive users of alternative products among young people aged 12 to 18 to highlight the difference between the groups.

## 2. Materials and Methods

### 2.1. Study Population

In 2017, Japan had 10,325 registered junior high schools and 4907 registered senior high schools in a total of 47 prefectures. For our study, schools were chosen for students to participate in a lifestyle survey of adolescents. The participating schools were selected using a cross-sectional random sample method with single-stage cluster sampling [18], wherein a school was considered a cluster unit. The sampling method included dividing Japan into regional blocks and randomly selecting schools from each block. The advantage of this methodology is the minimization of sampling bias [18]. Using the national school directory, junior high schools and senior high schools throughout Japan were randomly extracted, and the survey was administered to all students in these schools. Thereby, 98 of Japan’s junior high schools from 44 prefectures and 86 of the senior high schools from 42 prefectures were sampled; therefore, the selection rates were 0.95% and 1.75% of all schools, respectively. Private schools comprised 8.2% of the junior high schools and 19.8% of the senior high schools in the study. The surveys were administered from December 2017 through February 2018.

### 2.2. Data Collection

We approached the principal of each school for cooperation and sent the survey forms to the schools. The principals arranged for class teachers to distribute the forms to the students. The teachers explained to the students that participation was voluntary and that they should answer honestly. The students were given anonymous questionnaires and envelopes, which were completed and sealed by the students, collected by their teachers, and then returned to the research office with the seals intact.

### 2.3. Measures

The questionnaire included information about basic demographic data (sex, school grade, type of school); use of cigarettes, e-cigarettes and HTPs; exposure to secondhand smoke and understanding of the harmful effects of smoking; lifestyle behaviors and intentions regarding future education; and alcohol use. The list of the questions was provided in Appendix A.

### 2.4. Use of Cigarettes, e-Cigarettes and Heated Tobacco Products

#### 2.4.1. Discrimination of the Three Products

The three products were described in detail to ensure students were able to distinguish between them. Since we focused on investigating alternative products, we explained that a conventional cigarette is considered, ‘a cigarette made from rolled paper and tobacco and smoked with fire’. Due to the number of e-cigarettes currently for sale, we used the names of the most popular brands in the survey; for example, e-cigarettes included フレヴォ(FLEVO), エミリ (EMILI), ビタフル (VITAFUL) and ビタシグ (VITASIG). HTPs were also explained using product names to avoid any confusion; for example, heat-not-burn tobacco included アイコス (IQOS), プルームテック (Ploom Tech) and グロー (glo).

#### 2.4.2. Frequency of Use

To assess the frequency of conventional cigarette use, with the explanation of products above, we used two questions: ‘Have you ever smoked a conventional cigarette, including even a single puff?’ and ‘How many days have you smoked conventional cigarettes in the previous 30 days?’ Similar questions were used to determine the frequency of use of alternative products. For the experience question, ‘No, I have not’ or ‘Yes, I have’ were response options. To assess frequency of use, seven options were given: ‘0 days,’ ‘1–2 days,’ ‘3–5 days,’ ‘6–9 days,’ ‘10–19 days,’ ‘20–29 days,’ or ‘every day’.

‘Experience’ of cigarette/e-cigarette/HTP use was defined as smoking/using even once in the past; ‘current’ use of cigarettes/e-cigarettes/HTPs was defined as smoking/using at least once in the past 30 days. ‘Current use’ is more relevant to actual smoking behavior and is commonly used as an indicator of adolescent smoking. Furthermore, previous studies suggest that even minimal use of cigarettes leads to a significantly higher risk of becoming a chronic smoker [19,20]. Therefore, we determined ‘current use’ as a feasible outcome to use for analysis.

Additionally, in our analyses, we used ‘exclusive-use’ categories, meaning the exclusive use of a particular type of product. For instance, if we described ‘exclusive alternative product use’, the group included participants who currently used either e-cigarettes or HTPs but did not use conventional cigarettes. ‘Non-current user’ was defined as those who currently used neither cigarettes, e-cigarettes, nor HTPs. ‘Any conventional cigarette smokers’ were defined as those who currently smoke conventional tobacco cigarettes, including those who currently smoke conventional tobacco cigarettes exclusively and those who currently use conventional tobacco products plus HTP and/or e-cigarettes.

### 2.5. Lifestyle Behaviors and Intentions towards Future Education

In Japan, many students participate in after-school clubs. Some students engage in sports clubs at school, such as baseball, tennis or basketball; others choose cultural clubs, such as brass band, tea ceremony and flower arrangement. The questions asked about how often students participated in such ‘after school club’ activities as well as how often they had breakfast and their plans for the future. For analysis, answers indicating they eat breakfast ‘every day’ were categorized into ‘Yes’; answers of ‘sometimes’ or ‘seldom’ were categorized as ‘No.’ Regarding their plans for the future, students selected one out of seven options: ‘vocational school,’ ‘junior college,’ ‘college,’ ‘postgraduate school,’ ‘taking a job after graduating the current school,’ and ‘not decided yet’. We categorized those who selected ‘college’ or ‘postgraduate school’ into the ‘college or more’ group.

### 2.6. Exposure to Secondhand Smoke and Understanding the Harmful Effects of Smoking

For the survey questions about smoking exposure, we did not discriminate between the three products and described smoke as from ‘tobacco.’ In addition, ‘to smoke’ was the same as ‘to use tobacco.’ Students were asked about if and how often they were exposed to secondhand smoke at home. Participants who indicated they had been exposed at least once in the preceding seven days were categorized as ‘exposed.’ Survey questions were also used to determine how much students understood about the health risks of smoking. We categorized those who selected ‘I think that smoking is harmful’ into ‘Yes’ for understanding the harmful effects of smoking; all other responses were categorized as ‘No.’ Likewise, understanding the harm associated with secondhand smoking was evaluated; participants who selected ‘I think it is harmful’ were categorized into the ‘Yes’ group for understanding and all other responses were considered ‘No.’

### 2.7. Alcohol Use

We assessed how many days participants had engaged in drinking alcohol in the previous 30 days and the frequency of binge drinking. We consider drinking ‘a lot’ to be ‘five or more’ drinks of an ordinary can (350 mL) for beer and sweet cider. We defined a ‘current drinker’ as a student who had used alcohol on more than one or two days in the previous month and ‘binge drinkers’ as those who used alcohol including more than five cans of beer or sour at least once in a month. Specifically, having multiple drinks ‘once or twice in a month’, ‘once or twice in a week’, or ‘more than three times in a week’ were categorized as ‘yes’ for ‘binge drinking’ for the purpose of analysis.

### 2.8. Data Analysis

First, a descriptive analysis of the baseline characteristics of the study participants was performed including their grade levels (junior or senior high school), the overall prevalence of use of any products, and factors known to be associated with smoking. Second, Chi-squared tests were used to examine differences in each factor between any conventional cigarette smokers and non-current users. The Mantel–Haenszel test was used to compare the trends of the proportion of use in each school grade. Similarly, exclusive users of alternative products were compared with non-current users and then with any conventional cigarette smokers (including dual users and multiple users). For the latter comparison, we were interested in whether those youth who currently used only alternative products differed from those who currently smoked conventional cigarettes. Furthermore, in the comparison, we conducted the statistical tests for each of the 11 variables; hence, the Bonferroni correction was used to adjust the cut-off of *p*-values for significance due to multiplicity. Third, a multivariable logistic regression analysis was used to investigate the relationship between the use of cigarettes and several factors from the questionnaire. The adjusted odds ratio (OR) was calculated for each factor and its 95% confidence interval (95% CI) for cigarette use. Likewise, another logistic regression analysis was done with alternative product use. Finally, among any product users, the risk of alternative product use was compared with any conventional cigarette users in a third logistic regression model. Before the logistic regression analysis, the researchers discussed the relevance of all of the variables in this study and selected, by consensus, those most appropriate for inclusion in the models. Descriptive analysis and univariate analysis were performed using SPSS 25.0 (IBM Corp, New York, NY, USA). R i386 3.5.2 (R Foundation for Statistical Computing, Vienna, Austria) was used to conduct multivariable logistic regression analyses. Missing data were excluded from the analysis. We also conduct a supplementary analysis to compare exclusive cigarette users and exclusive alternative products (APs) users.

### 2.9. Ethical Statement

According to the Ethical Guidelines for Epidemiological Studies jointly announced by the Ministry of Health, Labor and Welfare and the Ministry of Education, Culture, Sports, Science and Technology of Japan, personal information is defined as follows: information of a living individual and the name, birthday and other descriptions included in that information that can be used to identify a specific individual. The questionnaire in our survey did not include any such information in consideration of identity protection and safeguarding privacy. This survey was reviewed and approved by the Ethics Review Committee of Tottori University School of Medicine when we conducted the survey (reference no. 17A078).

## 3. Results

The flow and results of the data collection are described in Figure 1. A total of 184 schools (98 junior high, 86 high school) were invited and 56.0% (103), including 49.0% of junior highs (48) and 64.0% of high schools (55), agreed to participate. The response rate of all students in the participating schools was 90.5% (64,152 of 70,927) for fully completed surveys, including 84.0% (22,215 of 26,604) from the junior highs and 94.6% (41,937 of 44,323) from the high schools.

### 3.1. Participant Characteristics

Table 1 shows the baseline characteristics of the study participants. In terms of lifestyles, the proportions of participants who indicated mostly healthy lifestyle habits was higher in junior high school students than senior high school students; however, more senior high students than junior high students intended to go to college or university. Regarding students who had used any products (tobacco or alternative): 4.1% in junior high and 7.3% in senior high school had at some time in their lives; 1.1% in junior high and 2.2% in senior high had at least once within the 30 days preceding the survey. The percentage of students who reported drinking alcohol at least once in the past month was 2.9% in junior high and 7.0% in senior high school. While binge drinking was quite rare in junior high school (0.6%), 1.9% of senior high school students responded that they had five or more cans of alcoholic beverages in one bout of drinking at least once in the preceding month.

To clarify the relationships between the use of the three different products, we created a Venn diagram (Figure 2) to show the total number of respondents who were ‘currently’ using each product. Overall, 1.8% (*n* = 1183) students reported using any product at least once in the prior month. Just over 40% were currently using more than one product; 200 students (17% of all current users) were ‘currently’ using all three products. Thirty percent of those using any product were exclusive e-cigarette or exclusive HTPs users, although compared to cigarettes and e-cigarettes, the number of respondents who only used HTPs was small. Thus, HTPs were most commonly used along with other products.

### 3.2. Comparison among Any Conventional Cigarette Users, Exclusive Alternative Products Users and Non-Users

Table 2 compares the proportion of respondents according to sociodemographic, lifestyle and other variables among three groups: people who do not currently use any products, any conventional cigarette users and exclusive AP users (the numbers vary slightly from table to table due to missing data). By Bonferroni correction, we adjusted the cut-off for significance as *p* < 0.0045. Overall, more males than females were users of any type of product; however, there was no significant difference between genders among users in terms of which type of product they used (cigarettes or APs). Regarding school grade, the trend in the proportions of the three groups were significantly different. For other variables—having breakfast every day, engaging in club activities, understanding the health effects of smoking, exposure to secondhand smoke and alcohol use—the proportions of students were significantly different between non-users and the other two groups. Moreover, the statistical tests showed that there were significant differences in those variables between AP users and any conventional cigarette users, except for that of having breakfast every day, future education intention and understanding of harmful effects of smoking.

### 3.3. The Association between Selected Factors and Any Conventional Cigarette Smoking or Exclusive Alternative Products Use

The results of a logistic regression analysis examining the risk factors for any conventional cigarette smokers compared with non-users are shown in Table 3. All factors—sex, school grade, understanding that smoking is harmful, having breakfast every day, participating in club activities, intending to go to college or a higher education course and present alcohol drinking—were significantly associated with any conventional cigarette smoking after mutual adjustment. Similarly, an analysis comparing lifestyle variables of exclusive AP users with those of non-users revealed that seven out of eight variables were significantly associated with AP use (Table 4). Although the students engaging in club activities were less likely to smoke cigarettes (OR 0.64, 95% CI 0.54, 0.76), the variable was not a significant preventive factor for exclusive AP use (OR 1.17, 95% CI 0.94, 1.48).

### 3.4. The Risk of Exclusive Alternative Product Use Compared with Any Conventional Cigarette Smoking among Any Product Users

Table 5 shows the result of logistic regression analysis which examined risk-associated lifestyle variables for exclusive AP users compared with those for any conventional cigarette smokers among users of any products. The results show that AP users were less likely to be higher school grade, more likely to participate in club activities (OR 1.61, 95% CI 1.22, 2.12), less likely to be exposed to secondhand smoking at home (OR 0.68, 95% CI 0.52, 0.90) and less likely to be current alcohol drinkers (OR 0.27, 95% CI 0.21, 0.35). In addition, there was a borderline significant association of exclusive AP users with ‘intention to pursue higher education’ (OR 1.35, 95% CI 0.99, 1.83).

The results of an additional logistic regression analysis were the same as for the groups compared above regarding club activities and alcohol use when comparing exclusive combustible cigarette smoking (conventional cigarettes only) to exclusive AP use, (Appendix A).

## 4. Discussion

The prevalence of any current product use in this Japanese sample, (1.1% of junior high school and 2.2% of senior high school) in 2017, was much lower than has been shown for students in the U.S. (7.2% of middle school and 27.1% of high school [10]) and UK (5% of 11–15 years old who smoke cigarettes at least once in a week and 6% of young people estimated as current e-cigarette users [21]). The prevalence of current cigarette smoking has continued to decline from 2000 to 2014; 9.4/5.6% (boys/girls) to 1.3/0.6% in junior high school students and 29.9/13.1% to 3.5/1.5% in high school students [6]. Our data indicates the continuous downward trends. The relationship between the three products illustrates a significant minority (30%) of those who used any products were exclusive AP users. The prevalence of smoking was much lower than alcohol use among Japanese adolescents.

We were interested in whether the factors related to exclusive use of APs are different from those of any conventional cigarette smokers. To examine this question, we compared three groups: people who do not currently use any products, current AP-exclusive users and any current conventional cigarette smokers. There were significant differences between non-users and exclusive APs users or any cigarette smokers (in all variables studied. In addition, comparing exclusive APs users and any cigarette smokers), four of the eight factors were statistically significant. These results suggest that participant characteristics may be different between non-users and any product users and some of these differed also between AP users and cigarette smokers. Across each variable, the results for AP users fell between the variable results for non-users and those for cigarette users.

The results of logistic regression analysis showed that known factors were significantly associated with any conventional cigarette smoking and APs use even after mutual adjustment. When exploring the difference between exclusive APs users and any conventional cigarette smokers—including those who also use HTPs or e-cigarettes, the conventional cigarette users were more likely to be exposed to secondhand smoke and drinking at home than AP-only users. Moreover, we found that there were differences in participating in club activities between the two groups; intention to pursue higher education also showed borderline significance. These results may suggest different factors—including social circumstances such as parental support, family income, or personal characteristics—may exist between exclusive AP users and those smoking cigarettes with or without the use of other products. Broadly consistent results were found when comparing exclusive cigarette smokes with exclusive AP users. The implications of this are unclear as those exclusively using APs may have previously smoked conventional cigarettes and stopped or may go on to smoke conventional cigarettes or remain exclusive AP users. This requires further research.

Our analytical strategy followed that of a previous study by Hanewinkel et al. that investigated risk factors associated with the use of e-cigarettes in a cohort study among German adolescents [22]. They compared the effect of each risk factor across e-cigarette, conventional cigarette and dual product use. The study implied the possibility that e-cigarettes attract a broader range of adolescents compared to conventional cigarettes. Several results of the present cross-sectional study were consistent with the Hanewinkel’s study. Gender and parent smoking showed significant effects on the use of both e-cigarettes and cigarettes. In addition, in their study, the relationship with a future academic career varied between e-cigarettes users and conventional cigarette users. Regarding HTPs, the findings of Wu et al.’s study indicate that relatively well-educated people tend to use HTPs [23]. These findings support our borderline significance in the association between education and exclusive APs use. Moreover, the previous cross-sectional study by Wills et al. [24] tried to determine whether established risk factors for smoking discriminate user categories by testing how e-cigarette users differed on a range of variables including social-cognitive factors, problem behavior risk factors and collateral substance use. Their findings showed parental factors, academic achievement, behavioral self-control, smoking expectancies, alcohol use, and heavy drinking significantly varied across non-users, e-cigarette exclusive users and dual users. Thus, their results also support our findings.

On the other hand, East et al. explained the former issue in a more nuanced way [25]. In their longitudinal study, they indicated that e-cigarette use was associated with cigarette use and vice versa. Certain psychological processes (‘common liabilities’) are used to explain the relationship of two. Specifically, curiosity, rebelliousness, and sensation-seeking were indicated as the psychological factors affecting product use. These psychological mechanisms also influence alcohol use. Hence, the strong association between any conventional cigarette smokers and alcohol use may be explained by these factors. In addition, the authors discussed several important differences to consider. In the UK, e-cigarettes are more accessible and convenient for young people compared to combustible cigarettes. Novel devices, the variety of flavors and the reduced risks of the new products have been demonstrated to be attractive to youth. Hair et al. also indicated that HTPs attract youth by the novelty [26]. Thus, several reports clarify that APs often appeal to those who are not smokers, but it remains unclear whether people in this group would have gone on to smoking if APs had not been on the market. Furthermore, it is necessary to further investigate the exclusive conventional cigarette use and the use of conventional cigarettes plus APs and their association with secondhand smoke at home, as there may be an implication that APs users intentionally avoid being exposed to and exposing others to secondhand smoke. On the other hand, it is possible that exclusive conventional cigarette users and those that use them along with APs may be high-risk groups who do not care about the negative consequences of exposure to multiple substances.

Our findings suggest several implications for future tobacco control. From the consistent results related to secondhand smoking at home, the smoking status of any family members significantly affects any type of product use among younger populations, suggesting that public health measures are needed to decrease adult smoking in order to prevent smoking among younger people and vice versa. Although the systematic reviews showed inconsistent results about the effect of e-cigarettes on smoking cessation, the latest PHE reports deduced that a considerable number of smokers quit after e-cigarettes were introduced in the UK [27]. Similarly, Lee et al. estimated that introducing a reduced-risk product into Japan substantially reduced smoking-related deaths [28]; further independent research would be useful in this area [29]. It is also worth mentioning that there is a gap in the implementation of MPOWER between Japan and the UK. As mentioned above, the regulation of tobacco, including APs, in Japan is unique and is behind global standards.

Furthermore, understanding of the harmful effects of smoking showed a protective effect against any type of product use. This finding implies that health education about smoking is an indispensable tobacco control measure. Adolescent trends in tobacco use have been decreasing and are at their lowest level seen in many years. We must maintain this trend of fewer adolescents using tobacco and keep up the use of this essential strategy—health education—with younger generations. It is also important to continue to monitor the trend of tobacco use including the novel alternative products in order to evaluate our current public health measures for tobacco control.

Our study includes several strengths. The data were collected from a nationwide large sample survey. This methodology enabled us to minimize sampling bias [18]. Hence, the result of this study can potentially be generalizable nationwide for Japanese adolescents. Although the proportion of current users of any of the three products was relatively small, the large sample size enabled us to select ‘current use of the products’ as the outcome for analysis, in contrast to previous studies that selected ‘having ever used’ as an outcome. However, it should be noted that our definition of ‘current use’ (in line with other research) is ‘any use within the last 30 days’ so it picks up a range of users including some who may only be using intermittently. In addition, Japan has a unique regulatory environment. E-cigarettes without nicotine are available, but HTP products have also been widely promoted since 2014. This means that the situation around cigarettes and APs differ from other countries. Given Japan’s unique situation, this study aimed to clarify the prevalence and risk factors of AP use among adolescents in Japan.

However, there are several limitations to the present study. First, the schools’ response rates were not as high as we expected, though the student-response rate was preserved, as it was high among those schools that did participate. Despite the efforts of the research team, ethical concerns and inconvenience due to the need for strict explanations may have caused the lower rate of cooperation among schools. However, the higher ethical concern was required to meet the criteria for recent ethical approval. Second, the fact that class teachers distributed the forms may have impacted the results. As stated above, to address students’ concerns about privacy, student questionnaires were anonymous, and the students put them into private envelopes themselves. In addition, the explanatory document given to the class teachers explained that they must ensure students’ privacy. Third, the consequences of multiple testing must be considered. As shown in Table 2, many hypotheses tests were performed, increasing the possibility of spuriously significant results. However, most of the significant results were lower than the adjusted *p* values—less than 0.004—and we factored this into our interpretation of the results. Fourth, the survey questions for the smoking environment only about asked about ‘tobacco’. This may have caused confusion for students answering questions as to if their parents used APs or about their exposure to aerosol from APs. Finally, as stated above, we should be cautious in interpreting the presented relationships identified in this cross-sectional analysis as causal. In addition, it is possible that confounding causes spurious associations, either through residual confounding of recorded variables, or variables that were not recorded at all. Therefore, present results are limited with regard to investigating what causes the relationship and accurate effect size. However, by using available variables, we tried to explore the factors which were associated with cigarettes and APs and compare them to clarify the relationships. As mentioned above, similar results from previous literature can support our findings.

## 5. Conclusions

In conclusion, the study results show that currently there is a very low prevalence of smoking and/or alternative products among youth in Japan aged 12 to 18 years. We found that the characteristics of alternative product users and any conventional cigarette smokers, differ from non-users, and there were some differences between exclusive AP users and any conventional cigarette smokers. Conventional factors consistently related to alternative products’ use indicate that reducing adult smoking and disseminating health education among adolescents remain important strategies for future tobacco control among adolescents. The priority must be to reduce tobacco use and nicotine addiction even further. To achieve the obvious goal, further research is necessary into the use of alternative products. Policymakers need to consider the updated measures on restrictions on labeling, advertising, sales to minors, pricing and taxation so that the Japanese national public health agenda goals can be achieved.

## Figures and Tables

**Figure 1 ijerph-17-03128-f001:**
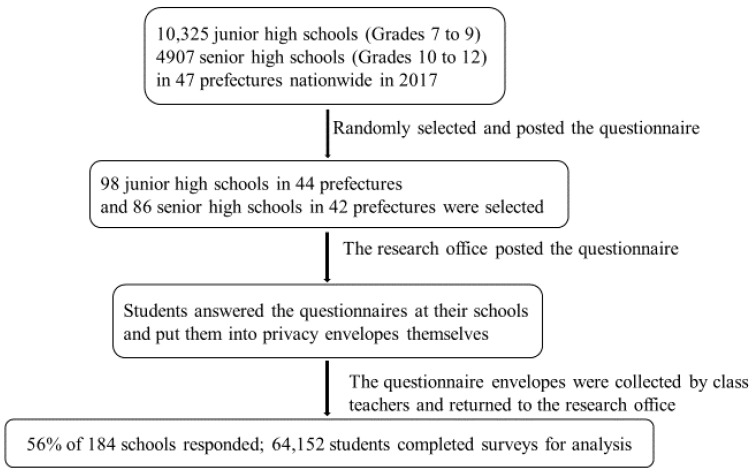
Flowchart of data collection.

**Figure 2 ijerph-17-03128-f002:**
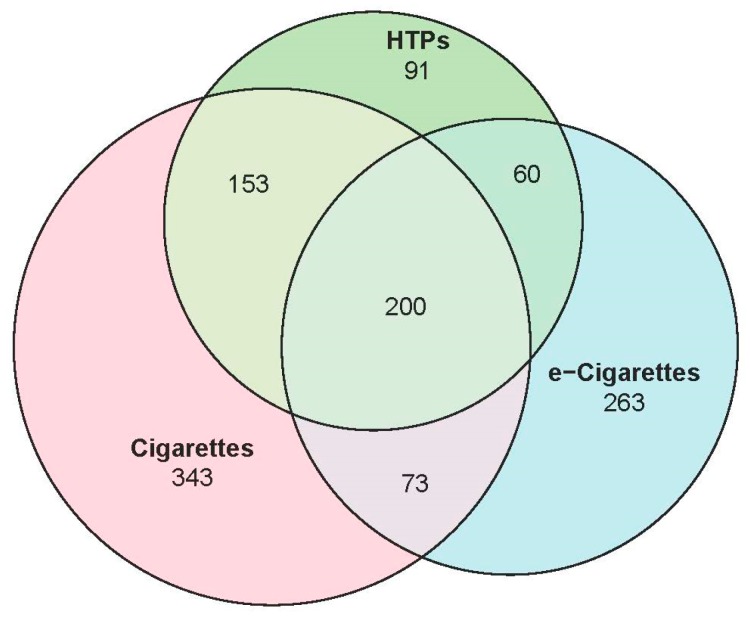
The number of current users of any products, Grade 7–12, male and female. In total, *n* = 1183 (1.8%; *N* = 64,152) The overlap areas represent those who used both or all; the non-overlap area indicates the total number of students who used each product exclusively. Abbreviations: HTPs = heated-tobacco-products.

**Table 1 ijerph-17-03128-t001:** Baseline characteristics of the study participants.

		Junior High School		Senior High School	
		(Grades 7 to 9)		(Grades 10 to 12)	
		*N* = 22,215		*N* = 41,937	
		*n*	(%)	*n*	(%)
Sex (Female)		11,036	(49.7)	18,534	(44.2)
School grade					
	First grade	7384	(33.2)	14,201	(33.9)
	Second grade	7329	(33.0)	14,212	(33.9)
	Third grade	7415	(33.4)	13,404	(32.0)
Having breakfast every day		19,079	(85.9)	34,183	(81.5)
Engaging in club activities		17,605	(79.3)	27,536	(65.6)
Future education intention (College or more)		4253	(19.1)	23,262	(55.5)
Experience of any tobacco or alternative products (Once in life)		911	(4.1)	3063	(7.3)
Current use of any tobacco or alternative products (Once in last 30 days)		244	(1.1)	939	(2.2)
Currently drinking alcohol (Once in last 30 days)		634	(2.9)	2950	(7.0)
Binge drinking ^a^		134	(0.6)	809	(1.9)

^a^ People who drink more than five cans of beer or sour at least once in a month.

**Table 2 ijerph-17-03128-t002:** Cross-comparison of three group: non-users with any conventional cigarette smokers and exclusive alternative-product use in youth, including demographics, lifestyles, school life and drinking habits.

		1. People Who Say They Do Not Use Any Products		2. Any Conventional Cigarette Users		3. Exclusive AP Users		1 vs. 2	1 vs. 3	2 vs. 3
		*N* = 62,969		*N* = 769		*N* = 414				
Variables		*n*	(%)	*n*	(%)	*n*	(%)	*p*-Value	*p*-Value	*p*-Value
Female gender		29,243	(46.4)	218	(28.3)	109	(26.3)	<0.001	<0.001	0.459
School grade ^a^								<0.001 ^b^	<0.001 ^b^	<0.001 ^b^
	Grade 7	7327	(99.2)	36	(0.5)	21	(0.3)			
	Grade 8	7246	(98.9)	38	(0.5)	45	(0.6)			
	Grade 9	7314	(98.6)	60	(0.8)	41	(0.6)			
	Grade 10	13,981	(98.5)	122	(0.9)	98	(0.7)			
	Grade 11	13,914	(97.9)	203	(1.4)	95	(0.7)			
	Grade 12	12,989	(96.9)	305	(2.3)	110	(0.8)			
Having breakfast every day		52,605	(83.5)	406	(52.8)	251	(60.6)	<0.001	<0.001	0.010
Engaging in club activities		44,558	(70.8)	330	(42.9)	253	(61.1)	<0.001	<0.001	<0.001
Future education intention (College or more)		27,251	(43.3)	155	(20.2)	109	(26.3)	<0.001	<0.001	0.015
Understanding that smoking is harmful		57,188	(90.8)	514	(66.8)	296	(71.5)	<0.001	<0.001	<0.100
Understanding that secondhand smoking is harmful		55,239	(87.7)	574	(74.6)	298	(72.0)	<0.001	<0.001	<0.321
Secondhand smoking at home		16,526	(26.2)	530	(68.9)	224	(54.1)	<0.001	<0.001	<0.001
Secondhand smoking out of home		18,576	(29.5)	621	(80.8)	225	(54.3)	<0.001	<0.001	<0.001
Currently drinking alcohol (Once in 30 days)		2884	(4.6)	546	(71.4)	154	(37.6)	<0.001	<0.001	<0.001
Binge drinking ^b^		583	(0.9)	316	(41.1)	44	(10.6)	<0.001	<0.001	<0.001

Abbreviations: APs = alternative products. Missing data were excluded in each analysis. *p*-Values are based on Chi-squared test. ^a.^ Mantel–Haenszel test for trend is used in the variable. ^b^ People who drink more than five cans of beer or cider at least once in a month.

**Table 3 ijerph-17-03128-t003:** Results of logistic regression: association between selected factors and any conventional cigarette smokers (*n* = 63,738).

Variables	OR	95% CI			*p*-Value
Sex					
Female	0.50	0.42	to	0.59	<0.01
Male (reference)	1.00				
School grade *	1.06	1.04	to	1.08	<0.01
Understand that smoking is harmful					
Yes	0.45	0.38	to	0.55	<0.01
No (reference)	1.00				
Having breakfast every day					
Everyday	0.69	0.58	to	0.83	<0.01
Sometimes, seldom (reference)	1.00				
Participating in club activities					
Yes	0.64	0.54	to	0.76	<0.01
No (reference)	1.00				
Intending to pursue higher education					
College or postgraduate school	0.49	0.40	to	0.59	<0.01
Others (reference)	1.00				
Secondhand smoking at home					
At least once in the preceding 7 days	3.18	2.68	to	3.78	<0.01
None in the preceding 7 days (reference)	1.00				
Currently drinking alcohol					
At least once in the previous month	34.66	29.31	to	41.12	<0.01
None in the previous month (reference)	1.00				

* This variable was modeled as a continuous variable. Abbreviations: OR = Odds ratio, 95% CI = 95% confidence interval.

**Table 4 ijerph-17-03128-t004:** Results of logistic regression: association between selected lifestyle variable factors and exclusive alternative products use (*n* = 63,383).

Variables	OR	95% CI			*p*-Value
Sex					
Female	0.48	0.38	to	0.60	<0.01
Male (reference)	1.00				
School grade *	1.03	1.01	to	1.06	<0.01
Understand that smoking is harmful					
Yes	0.47	0.37	to	0.60	0.01
No (reference)	1.00				
Having breakfast every day					
Everyday	0.59	0.47	to	0.75	<0.01
Sometimes, seldom (reference)	1.00				
Participating in club activities					
Yes	1.17	0.94	to	1.48	0.17
No (reference)	1.00				
Intending to pursue higher education					
College or postgraduate school	0.61	0.48	to	0.77	<0.01
Others (reference)	1.00				
Secondhand smoking at home					
At least once in the preceding 7 days	2.24	1.82	to	2.76	<0.01
None in the preceding 7 days (reference)	1.00				
Currently drinking alcohol					
At least once in the previous month	9.29	7.51	to	11.46	<0.01
None in the previous month (reference)	1.00				

* This variable was modeled as a continuous variable. Abbreviations: OR = odds ratio, 95% CI = 95% confidence interval.

**Table 5 ijerph-17-03128-t005:** The risk of exclusive alternative product use compared with any conventional cigarette use among any product users (*n* = 1183).

Variables	OR	95% CI			*p*-Value
Sex					
Female	1.02	0.75	to	1.37	0.91
Male (reference)	1.00				
School grade *	0.96	0.93	to	0.99	0.02
Understand that smoking is harmful					
Yes	1.18	0.89	to	1.57	0.26
No (reference)	1.00				
Having breakfast everyday					
Everyday	1.03	0.78	to	1.35	0.84
Sometimes, seldom (reference)	1.00				
Participating in club activities					
Yes	1.61	1.22	to	2.12	<0.01
No (reference)	1.00				
Intending to pursue higher education					
College or postgraduate school	1.35	0.99	to	1.83	0.06
Others (reference)	1.00				
Secondhand smoking at home					
At least once in the preceding 7 days	0.68	0.52	to	0.90	0.01
None in the preceding 7 days (reference)	1.00				
Currently drinking alcohol					
At least once in the previous month	0.27	0.21	to	0.35	<0.01
None in the previous month (reference)	1.00				

* This variable was modeled as a continuous variable. Abbreviations: OR = Odds ratio, 95% CI = 95% confidence interval. An odds ratio of greater than one indicates that exclusive alternative product users had higher odds of the factor than any conventional cigarette users.

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
