# Peer review of "Comparing Factors Related to Any Conventional Cigarette Smokers, Exclusive New Alternative Product Users, and Non-Users among Japanese Youth: A Nationwide Survey"

_ijerph, 2020, doi:10.3390/ijerph17093128_

Round 1
Reviewer 1 Report
This is an interesting study on the hot topic of youth tobacco product use. The authors have taken several steps to ensure that the design of this type of study is optimal and collected detailed information on tobacco use. However, there are several areas that the study needs improvement including conceptualization of the analytical elements, the analysis and the interpretation. I highlight some of these areas below:
The title is not reflecting the analyses within the manuscript. The comparisons (of the logistic models) were between non-users and users in addition to cigarette users and AP users. Please modify to better reflect what was compared in the study.
What was the Response rate of students?
“Current Cigarette user” is not clearly defined. It seems that the authors really mean poly-tobacco product users. If that is the case then they should justify why comparing polyuse with AP use is meaningful. I also suggest they compare exclusive combustible cigarette use with exclusive AP use separately.
The authors state that “ independent variables in the logistic regression model were selected considering the correlation between each variable.” Why? I suggest that the authors use all available information to adjust their models. This is an observational study, so the more information one includes to adjust their model the more unbiased the results.
Can authors adjust for multiplicity? There more than one comparison and the p-values should be adjusted accordingly. The authors acknowledge that but they don’t correct for it. A simple Bonferroni correction would work.
Please, state clearly in the logistic regression tables which is the reference category.
In the discussion the author state that “We were interested in whether Aps attract young people who are less likely to become conventional cigarette smokers”. This cannot be evaluated with the data presented. This will require time-dependent data. This is a correlational study. Please modify this statement to reflect associations.
The main message of the paper seems to be that poly-users are more likely to be exposed to SHS and drinking than AP-only use. This is hardly a novel finding. However, it replicates previous findings pointing to negative exposures of poly-users compared to AP users. The discussion should focus largely on that and integrate previous studies that similarly compared poly-use and AP product users such as e-cig users.
Author Response
Reviewer Comments:
Reviewer 1
General comment: This is an interesting study on the hot topic of youth tobacco product use. The authors have taken several steps to ensure that the design of this type of study is optimal and collected detailed information on tobacco use. However, there are several areas that the study needs improvement including conceptualization of the analytical elements, the analysis and the interpretation. I highlight some of these areas below.
General Response: We appreciate the time and effort you have dedicated to providing constructive feedback on our manuscript.
Point 1: The title is not reflecting the analyses within the manuscript. The comparisons (of the logistic models) were between non-users and users in addition to cigarette users and AP users. Please modify to better reflect what was compared in the study.
Response 1: We agree with your assessment that the title does not reflect what we examined. We have revised the title (lines 2–4) in order to help reader to follow the aim of our study. The new title is:
‘Comparing factors related to any conventional cigarette smokers, exclusive new alternative product users, and non-users among Japanese youth: A nationwide survey’
Point 2: What was the Response rate of students?
Response 2: Thank you for raising this important point. We have added the response rate of students in the Results section (Results section, lines 223–227):
“A total of 184 schools (98 junior high, 86 high school) were invited and 56.0% (103), including 49.0% of junior highs (48) and 64.0% of high schools (55), agreed to participate. The response rate of all students in the participating schools was 90.5% (64,152 of 70,927) for fully completed surveys, including 84.0% (22,215 of 26,604) from the junior highs and 94.6% (41,937 of 44,323) from the high schools.”
Point 3: ‘Current Cigarette user’ is not clearly defined. It seems that the authors really mean poly-tobacco product users. If that is the case then they should justify why comparing polyuse with AP use is meaningful.
Response 3: Thank you for pointing this out.
In our study, ‘conventional cigarette users’ meant any conventional cigarette users (who may or may not also use HTPs and/or e-cigarettes). Also, e-cigarettes do not contain tobacco, which is what differentiates them from heat-not-burn tobacco products (HTPs) and hence we would disagree in referring to the use of these in any definition of poly-tobacco product uses. The definition provided in section 2.4.2, lines 158–160 was revised to better clarify that definition and the term was changed to ‘any conventional cigarette smokers’, inclusive of exclusive conventional cigarette smokers and those who also use either or both of the other product types, the APs (alternative products). See new text added on lines 158-160:
‘Any conventional cigarette smokers’ were defined as those who currently smoke conventional tobacco cigarettes, including those who currently smoke conventional tobacco cigarettes exclusively and those who currently use conventional tobacco products plus HTP and/or e-cigarettes.’
We believe that this comparison is meaningful because we were interested in whether those people who only used alternative products differed from those who smoke cigarettes. The limitation is that those who are using alternative products may be those who had smoked conventional cigarettes and stopped (as they were aware they were less harmful) or they may go on to smoke conventional cigarettes. Nevertheless, if they were a completely different group than those smoking conventional cigarettes, it could suggest the presence of alternative products was drawing a new group of users to these products.
Point 4: I also suggest they compare exclusive combustible cigarette use with exclusive AP use separately.
Response 4: You have raised an important suggestion. We have provided the results of logistic regression analysis which compared exclusive combustible cigarette use with exclusive APs use in Supplementary file 2. We also have discussed the finding in Results (Results section, lines 312–315) and Discussion (Discussion section, lines 335–347).
Point 5: The authors state that ‘independent variables in the logistic regression model were selected considering the correlation between each variable.’ Why? I suggest that the authors use all available information to adjust their models. This is an observational study, so the more information one includes to adjust their model the more unbiased the results.
Response 5: We agree with your assessment regarding the logistic regression model. To help our analysis strategy read more clearly, we have rewritten these sentences. In the revised text, we have written how authors select the variables. Before the logistic regression analysis, we discussed which variables were relevant to this study and used consensus to develop the strategy. (Materials and Methods section, lines 207–209). No multicollinearity was detected in any logistic regression model conducted.
Point 6: Can authors adjust for multiplicity? There more than one comparison and the p-values should be adjusted accordingly. The authors acknowledge that but they don’t correct for it. A simple Bonferroni correction would work.
Response 6: You have raised an important suggestion. We have incorporated your suggestion by using the Bonferroni correction in multiple tests. We have revised our manuscript. (Materials and Methods section, lines 199–201; Results section, lines 262; and Discussion section, lines 421–424).
Point 7: Please, state clearly in the logistic regression tables which is the reference category.
Response 7: We agree with your assessment regarding the logistic regression tables. We have incorporated your suggestions in Table 3 to 5.
Point 8: In the discussion the author state that ‘We were interested in whether Aps attract young people who are less likely to become conventional cigarette smokers’. This cannot be evaluated with the data presented. This will require time-dependent data. This is a correlational study. Please modify this statement to reflect associations.
Response 8: We agree with your assessment regarding the expression in the manuscript. We have modified the sentence with your suggestion. (Discussion section, lines326–327):
“We were interested in whether the factors related to exclusive use of APs are different from those of any conventional cigarette smokers.”
Point 9: The main message of the paper seems to be that poly-users are more likely to be exposed to SHS and drinking than AP-only use. This is hardly a novel finding. However, it replicates previous findings pointing to negative exposures of poly-users compared to AP users.
Response 9: You have raised an important suggestion. We agree with your suggestion that our findings replicated previous findings in terms of e-cigarettes. However, we are convinced that our research results are also novel in that this is the first study that examined the association among Japanese adolescents. Moreover, it is quite unique that HTPs are used among adolescents under the unique regulation in Japan.
Point 10: The discussion should focus largely on that and integrate previous studies that similarly compared poly-use and AP product users such as e-cig users.
Response 10: We agree with your assessment regarding the discussion. General discussion of the comparison of results with prior studies is given in lines 348–377.We have rechecked if the use of products were clearly described in the discussion with the prior literature.
Reviewer 2 Report
Dear Authors,
your paper presents interesting results from a large study. You have prepared nice manuscript and I have only few comments
Methods sections:
Could you describe how did you manage missing data?
Can you you explain for international audience what "club activities" mean?
results
tab3 and tab4 has used to terms "currently drinking alcohol" and "presently drinking alcohol'. I thing you should unify wording.
discussion
line 325 - Why does term used in question "smoking near you at home" equals here to "parent smoking"? Are you sure they are the same?
line 326 "a future" not "e future"
line 349 correct word "iadult"
weaknesses:
Did the fact that class teachers distributed the forms could have impact on results?
You have discussed low response rate at school level. How was with response rate inside schools?
conclusion
line 404 replace 'our" with "Japanese"?
Author Response
Reviewer Comments:
General comment: Your paper presents interesting results from a large study. You have prepared nice manuscript and I have only few comments.
General Response: We appreciate the time and effort you have dedicated to providing constructive feedback on our manuscript.
Methods section
Methods Point 1: Could you describe how did you manage missing data?
Response 1: Thank you for raising this important point. Missing data were excluded from the analysis because we determined that the exclusion would not skew the results. We added some text to clarify this in our manuscript (Material and Methods section 2.8, lines 210–211): “Missing data were excluded from the analysis”
Methods Point 2: Can you explain for international audience what ‘club activities’ mean?
Response 2: You have raised an important suggestion. We have explained this in the Methods section (Material and Methods section, lines 162–165):
“In Japan, many students participate in after-school clubs. Some students engage in sports clubs at school, such as baseball, tennis, or basketball; others choose cultural clubs, such as brass band, tea ceremony, and flower arrangement. The questions asked about how often students participated in such ‘after school club’ activities as well as how often they had breakfast and their plans for the future.”
Results section
Results Point 3: Tab3 and tab4 has used to terms ‘currently drinking alcohol’ and ‘presently drinking alcohol'. I thing you should unify wording.
Response 3: Thank you for pointing this out, we have rewritten and unify wording in the tables to read “currently drinking alcohol”.
Discussion section
Discussion Point 4: Line 325 - Why does term used in question ‘smoking near you at home’ equals here to ‘parent smoking’? Are you sure they are the same?
Response 4: We agree with your assessment regarding the second-hand smoking at home in the manuscript. We should recognize this means second-hand smoking from any family members at home. We have rewritten the sentences and used the consistent term. (Discussion section, line 376–381, 383).
“Furthermore, it is necessary to further investigate the exclusive conventional cigarette use and the use of conventional cigarettes plus APs and their association with second-hand smoke at home, as there may be an implication that APs users intentionally avoid being exposed to and exposing others to second-hand smoke. On the other hand, it is possible that exclusive conventional cigarette users and those that use them along with APs may be high-risk groups who do not care about the negative consequences of exposure to multiple substances. “
Discussion Point 5: Line 326 ‘a future’ not ‘e future’
Response 5: We apologize for our carelessness. We have revised (Discussion section, line 354).
Discussion Point 6: Line 349 correct word ‘iadult’
Response 6: We apologize our carelessness. We have revised (Discussion section, line 385).
Discussion Point 7: Did the fact that class teachers distributed the forms could have impact on results?
Response 7: You have raised an important suggestion. We have paid attention to avoid such impact on our survey. As we described in 2.2 Data Collection, students answer anonymous questionnaires and put them into private envelopes themselves. Also, the explanatory document for class teachers explained that they must ensure students’ privacy. We have added the discussion about this issue (Discussion section, lines 417–421).
Discussion Point 8: You have discussed low response rate at school level. How was with response rate inside schools?
Response 8: Thank you for raising this important point. We have added the response rate of students in the Results section (Results section, lines 223–227). Also, we address the student response rate in the Discussion section (Discussion section, lines 413–414).
Conclusion section
Conclusion Point 9: Line 404 replace ‘our’ with ‘Japanese’?
Response 9: Thank you for your suggestion. We have revised this point. (Conclusion section, line 444)